# Transcriptional Profiling of Aflatoxin B1-Induced Oxidative Stress and Inflammatory Response in Macrophages

**DOI:** 10.3390/toxins13060401

**Published:** 2021-06-04

**Authors:** Jinglin Ma, Yanrong Liu, Yongpeng Guo, Qiugang Ma, Cheng Ji, Lihong Zhao

**Affiliations:** State Key Laboratory of Animal Nutrition, College of Animal Science and Technology, China Agricultural University, Beijing 100193, China; majinglin@cau.edu.cn (J.M.); liuyanrong@cau.edu.cn (Y.L.); guoyongpeng@cau.edu.cn (Y.G.); maqiugang@cau.edu.cn (Q.M.); jicheng@cau.edu.cn (C.J.)

**Keywords:** transcriptional profiling, aflatoxin B1, oxidative stress, inflammatory response, macrophages

## Abstract

Aflatoxin B1 (AFB1) is a highly toxic mycotoxin that causes severe suppression of the immune system of humans and animals, as well as enhances reactive oxygen species (ROS) formation, causing oxidative damage. However, the mechanisms underlying the ROS formation and immunotoxicity of AFB1 are poorly understood. This study used the mouse macrophage RAW264.7 cell line and whole-transcriptome sequencing (RNA-Seq) technology to address this knowledge-gap. The results show that AFB1 induced the decrease of cell viability in a dose- and time-dependent manner. AFB1 also significantly increased intracellular productions of ROS and malondialdehyde and decreased glutathione levels. These changes correlated with increased mRNA expression of NOS2, TNF-α and CXCL2 and decreased expression of CD86. In total, 783 differentially expressed genes (DEGs) were identified via RNA-Seq technology. KEGG analysis of the oxidative phosphorylation pathway revealed that mRNA levels of ND1, ND2, ND3, ND4, ND4L, ND5, ND6, Cyt b, COX2, ATPeF0A and ATPeF08 were higher in AFB1-treated cells than control cells, whereas 14 DEGs were downregulated in the AFB1 group. Furthermore, seven immune regulatory pathways mediated by oxidative stress were identified by KEGG analysis. Altogether, these data suggest that AFB1 induces oxidative stress in macrophages via affecting the respiratory chain, which leads to the activation of several signaling pathways related to the inflammatory response.

## 1. Introduction

Aflatoxins (AFT) are a group of secondary metabolites mainly produced by *Aspergillus flavus* and *Aspergillus parasiticus*. They are found in various agricultural commodities and lead to large annual losses in the agricultural industry [1,2]. At present, more than 20 kinds of AFT have been isolated and chemically characterized, e.g., aflatoxin B1 (AFB1), aflatoxin G1 (AFG1) and aflatoxin M1 (AFM1) [3,4]. Among the AFT identified, AFB1 is the predominant form that presents the highest toxic potential [5]. As a potent carcinogen, AFB1 is also a major contributor to the worldwide occurrence of hepatocarcinoma, with about 4.6–28.2% of liver cancer cases linked to AFB1 exposure [6]. This association led the International Agency for Research on Cancer to categorize AFB1 as class I carcinogen [7]. In addition to its role in hepatocellular carcinoma, AFB1 can also cause other serious health problems including mutagenesis [8,9], growth retardation [10,11] and immune suppression [12,13]. 

AFB1’s high toxicity is thought to be due its metabolite (AFB1-8,9-epoxide) that can widely and irreversibly react with guanine residues to generate highly mutagenic DNA adducts [4,14]. In addition, AFB1 toxicity is also related to its strong induction of oxidative stress, and the formation of reactive oxygen species (ROS) is one of the major triggers of its toxic mechanism [15,16]. The overproduction of ROS can directly cause chemical damage to DNA, proteins and lipids, which can trigger apoptosis or transform cells into malignancy [17,18]. Paradoxically, ROS generation induced by AFB1 can activate immune response. Previous studies have indicated that AFB1 significantly increases autophagy of macrophages from mice and humans by activating the ROS-mediated MEK/ERK signaling pathway [19]. Studies in our laboratory using broiler chickens fed AFB1 contaminated diets revealed upregulated expression of pro-inflammatory cytokines through ROS-mediated nuclear factor kappa B (NF-κB) signaling pathway in the livers of AFB1 animals [20].

In fact, the impact of AFB1 on the immune system has long been observed by clinicians in humans and animals. The AFB1 induced immune dysregulation may lead to immunosuppression resulting decreased resistance to infectious diseases and reduced vaccine and therapeutic efficacy in the host [21,22]. The inflammatory response is a key component of immune response to many types of pathogens [23]. However, it is widely recognized that inflammation also plays an essential role in tumor progression and metastasis. In the tumor microenvironment, various inflammatory mediators, such as cytokines, chemokines, exosomes and reactive oxygen species constituents, are continuously produced. These mediators represent a critical interface between immune and neoplastic compartments. They not only continuously support tumor survival and expansion but also suppress antitumor adaptive immune responses [24]. Several published studies have provided evidence that AFB1 hepatoxicity is associated with inflammation [25]. There are two ways AFB1 affect the inflammatory response. First, AFB1 can directly affect the viability of phagocytes (dendritic cells, neutrophils and macrophages) which are a fundamental part of inflammatory immunity [26,27]. Meanwhile, some studies have shown that AFB1 exposure inhibits in vitro phagocytosis of macrophages [28,29]. More importantly, AFB1 can also modify the synthesis of inflammatory cytokines, which has been found by in vitro studies using human [30] and mouse [31] cells. 

While some reports demonstrate that AFB1 can induce oxidative stress in various cells, the molecular mechanism behind it remains unclear. Furthermore, most previous studies only focus on one signaling pathway. The role of other signaling pathways and how they may interact with each other is unclear. Nevertheless, to obtain new insights into the oxidative stress induced by AFB1 and the immunotoxicity of AFB1, we used whole-transcriptome sequencing (RNA-Seq) technologies to investigate the effects of AFB1 on mitochondrial oxidative phosphorylation and inflammatory response. To our best knowledge, this is the first study that included RNA-Seq technology in the experimental design to investigate the immunotoxicity of AFB1 on macrophages. Our results illustrate that AFB1 induces oxidative stress in macrophages via affecting the respiratory chain, which leads to the activation of several signaling pathways related to the inflammatory response, complementing the various immune function studies on AFB1 that have been reported. 

## 2. Results

### 2.1. Toxic Effects of AFB1 on Cell Viability

The cell viability was measured, as shown in Figure 1. As the concentration of AFB1 increased from 0 to 100 μM, the viability of the RAW264.7 cells decreased, and this decrease was greater the longer the cells were exposed. After 24 h of exposure to 50 μM AFB1, the cell viability began to significantly decrease compared with the control group (*p* < 0.05), and the cell viability decreased by 30.22% (*p* < 0.001) when treated with 100 μM AFB1 for 24 h. When the treatment time increased to 48 h, a significant difference in cell viability was first observed at 12.5 μM AFB1 (*p* < 0.01), with the maximum observed difference in cell viability being 38.67% (*p* < 0.001) when treated with 100 μM AFB1. Based on these data, 25, 50 and 100 μM AFB1 were chosen for subsequent experiments.

### 2.2. Effect of AFB1 on ROS Production in RAW264.7 Cells

Confocal laser scanning microscopy was used to detect AFB1 induced ROS using DCFH-DA fluorescence (Figure 2a). As the amount of AFB1 increased, we observed an increase in fluorescence intensities. Then, the level of the fluorescence was quantified using Image J software, as shown in Figure 2b. This analysis demonstrates all three doses of AFB1 stimulated a significant increase in intracellular ROS levels. Compared with the control group, the ROS level was more than doubled in the 25 μM AFB1 group (*p* < 0.001), and it was more than tripled in the 50 and 100 μM AFB1 groups (*p* < 0.001).

### 2.3. Effect of AFB1 on GSH and MDA Contents in RAW264.7 Cells

The concentrations of GSH and MDA were measured, as shown in Figure 3. In Figure 3a, a dose-dependent reduction in GSH content was observed in cells treated with 25, 50 and 100 μM AFB1. No matter the treatment time, compared with the control group, the levels of GSH in the three AFB1 groups were significantly decreased (*p* < 0.01), which markedly decreased by 24.88% (24 h) and 25.31% (48 h) at the concentration of 100 μM AFB1 (*p* < 0.001). By contrast, we could see an increasing trend in MDA content with an increasing concentration of AFB1 (Figure 3b). After being exposed to AFB1 for 24 h, we noticed that 25 μM AFB1 did not affect the intracellular MDA content (*p* > 0.05), whereas. at 50 and 100 μM AFB1, the MDA contents were significantly increased (*p* < 0.05). After 48 h of exposure to AFB1, the levels of MDA in the three AFB1 groups were all significantly increased (*p* < 0.05), which nearly tripled from 2.75 to 7.75 μmol/g protein at the concentration of 100 μM AFB1 (*p* < 0.001). Based on these data, 50 μM AFB1 and 24 h were chosen for subsequent research.

### 2.4. Effect of AFB1 Exposure on Gene Expression of Inflammatory Cytokines

The expression levels of nine genes related to inflammation response were measured by real-time RT-PCR, as shown in Figure 4. These data show that cells treated with 50 μM AFB1 for 24 h responded with a significant increase in the expression of pro-inflammatory factors NOS2, TNF-α and CXCL2 (*p* < 0.05), but not IL-6 (*p* > 0.05). AFB1 treatment did not have a significant influence on the production of ARG1, TGF-β and IL-10 (*p* > 0.05). Additionally, we assayed for changes in the expression levels of macrophage surface markers CD86 and CD206. The results indicate that 50 μM AFB1 noticeably decreased the expression of CD86 in macrophage cells (*p* < 0.01), but there was no significant difference in the expression of CD206 between the cells not exposed to AFB1 and the cells exposed to 50 μM AFB1 (*p* > 0.05).

### 2.5. Transcriptomic Analysis of the Effects of AFB1 on RAW264.7 Cells

Samples yielded sufficient transcripts and mapped to 24,859 *Mus musculus* genes. DEGs between the control and AFB1 groups were defined by a magnitude fold change ≥ 2 and −log10 (*p*-value) ≤ 0.05. Using these criteria, DEGs were identified, as shown in Figure 5a. In total, 1124 DEGs were identified between control and AFB1 treated cells. As shown in Figure 5a, compared with the control group, AFB1 exposure induced a significant upregulation of 240 genes in RAW264.7 cells, whereas 884 DEGs had lower expression in the AFB1 group as compared with the control group. The heat map (Figure 5b) shows an unsupervised hierarchical clustering of samples. There was a clear distinction between the two groups, which indicates that the AFB1 group had a distinguishable and reproducible change in gene expression patterns compared to the control group.

KEGG pathway enrichment analysis of the DEGs allowed the genes to be categorized based on enriched biological regulation processes. Figure 5c illustrates the top 30 KEGG pathways, ranked by *p*-value. The biological processes “oxidative phosphorylation”, “ribosome”, “cardiac muscle contraction”, “biosynthesis of amino acids”, “glycine, serine and threonine metabolism” and “folate biosynthesis”, among others, were significantly annotated with KEGG pathway (*p* < 0.05). The most significant KEGG pathway was “oxidative phosphorylation” (*p* = 5.13 × 10^−16^), and 31 DEGs were categorized on it with the highest percentage of genes per KEGG at 11.57%. 

### 2.6. KEGG Analysis of the Effect of AFB1 on Oxidative Phosphorylation Pathway

The DEGs categorized in “oxidative phosphorylation” are listed in Figure 6. The pathway maps the DEGs across four multi-enzymatic respiratory complexes (Complexes I–IV) and ATP synthase, which are embedded in the inner mitochondrial membrane. In Complex I, we found that six genes (ND1, ND3, ND4, ND4L, ND5 and ND6) belonging to the ND gene family were upregulated (*p* < 0.05) and seven genes (Ndufs3, Ndufv1, Ndufa2, Ndufa7, Ndufa9, Ndufb6 and Ndufb7) of the Nduf gene family were downregulated in the AFB1 group compared to control group (*p* < 0.05). In Complex II, the mRNA levels of SDHB and SDHD were downregulated after AFB1 exposure. In Complex III, Cyt1, ISP and QCR8 expressions in the AFB1 group were lower than those in the control group (*p* < 0.05), whereas the expression level of Cyt b was higher in the AFB1 group (*p* < 0.05). In Complex III, COX2 was significantly upregulated (*p* < 0.05), while COX5A, COX6C and COX11 were downregulated (*p* < 0.05). Finally, expression of AFTeF1E in the F0 unit of ATP synthase (*p* < 0.05) was decreased and ATPeF0A and ATPeF08 in the F1 unit of ATP synthase increased (*p* < 0.05) after treated with 50 μM AFB1 for 24 h.

### 2.7. Relative Signaling Pathways Mediated by AFB1-Induced Oxidative Stress

To validate the differences in gene expression induced by AFB1 treatment by RT-qPCR analysis, the expression levels of the inflammation associated genes IL-1β, IL-6, IL-10, CXCL2, NOS2, ARG1, TGF-β, TNF-α, CD80, CD86, CD163 and CD206, as detected by RNA-Seq analysis, were compared to results from RT-qPCR (Table 1). The results are consistent with data shown in the RT-qPCR experiment, further confirming that AFB1 induced an inflammatory response by affecting ROS production.

Due to the significant “oxidative phosphorylation” pathway enrichment, further assessment of oxidative stress-related immune regulatory pathways was conducted, as shown in Table 2. These pathways included, among others, peroxisome proliferator-activated receptor (PPAR) signaling pathway, mammalian target of rapamycin (mTOR) signaling pathway, nuclear factor-kappa B (NF-κB) signaling pathway, phosphatidylinositol 3 kinase-Akt (PI3K-Akt) signaling pathway, janus kinases-signal transducer and activator of transcription proteins (JAK-STAT) signaling pathway, mitogen-activated protein kinase (MAPK) signaling pathway and tumor necrosis factor receptor (TNF) signaling pathway. Among the listed pathways, PPAR signaling pathway had the lowest KEGG pathway P-value (*p* = 0.17). However, the PI3K-Akt signaling pathway contained the greatest number of DEGs, with 25 DEGs annotated on this pathway.

## 3. Discussion

Macrophages are important innate immune cells that originate from progenitors in the bone marrow and can be found throughout the body [32,33]. Apart from maintaining tissue homeostasis, they also play a critical role in initiating innate immune responses and directing subsequent acquired immune response [34]. Many results obtained from in vivo and in vitro studies indicate that AFB1 can decrease the viabilities of several kinds of immune cells, as well as impair their functions [35,36,37]. In this experiment, RAW264.7 cells were treated with 0, 3.125, 6.25, 12.5, 25, 50 or 100 μM AFB1 for 24 and 48 h, respectively, and the results show a similar dose- and time-dependent decrease in cell viability, indicating that AFB1 negatively affect the RAW264.7 cells. One proposed mechanism for AFB1 toxicity is that metabolites of AFB1 can lead to DNA damage, thus promoting apoptosis [4]. Another theory is oxidative damage. In fact, a recent study has shown that AFB1 activates mitochondrial ROS-dependent signaling pathways which induce apoptosis [18]. 

Oxidative stress has been defined as an imbalance between the intracellular production of ROS and the antioxidant system. When the generation of free radicals exceeds the antioxidant capacity of a cell, oxidative stress occurs in the cell [38,39]. Actually, many investigators have observed overproduction of ROS in macrophages exposed to AFB1 [40,41]. Likewise, our results show that the level of ROS produced by RAW264.7 macrophage cells in the AFB1 treatment group (25, 50 and 100 μM AFB1) was significantly higher than the control group, which is consistent with previous studies. Collectively, these results demonstrate that oxidative stress plays a major role in the toxic effects of AFB1, especially its immunotoxicity. The increasing level of ROS can induce lipid peroxidation by attacking cellular and organelle membranes, leading to the production of MDA [42]. As shown in our experiment, the level of MDA in RAW264.7 cells was increased after exposure to AFB1, indicating that oxidative stress does exist in macrophage cells. In general, these results are in agreement with the previous studies of AFB1 toxicity [43,44]. Through cooperating with primary and secondary antioxidant enzymes, such as glutathione reductase, glutathione peroxidase and glutathione S-transferase, GSH plays an important role in the detoxification of intracellular ROS [45]. As ROS levels increase, GSH will be oxidized to glutathione disulfide (GSSG). Therefore, the lower level of GSH in the AFB1 treatment group in our study confirms that AFB1 induced oxidative stress in RAW264.7 cells. These findings are in line with the result that dose-dependent decreases in GSH levels were observed after incubation with AFB1 [46,47].

The ability of AFB1 to modulate cytokines expression has also been established in both humans and animal models. However, the results obtained from different studies might lead to conflicting conclusions. Some studies reported that AFB1 can induce M1-like polarization in macrophage cells, which was represented by higher levels of TNF-α, IL-6 and CD86 mRNA and a lower level of CD206 mRNA production after AFB1 treatment [19,46]. However, other studies reported that 50 μM AFB1 exposure decreased the expression of pro-inflammatory factors, including IL-1, IL-6 and TNF-α in macrophage cells [30,31,48], indicating that AFB1 induced M2-like polarization in macrophages. What causes the inconsistent trends of results might be the difference in the amount of AFB1 and the duration of exposure used in these studies. A recently study demonstrated that AFB1 could induce macrophages to display M1 phenotype firstly, but later induce a switch to the M2 phenotype [49]. Therefore, differences in the pro-inflammatory and anti-inflammatory cytokines could be a matter of how long post AFB1 exposure. Additionally, macrophages used in these experiments were generally derived from different species and different organs. The distinction of their toleration to AFB1 might also be the reason cytokines expression was different even though the amount of AFB1 and the duration of exposure were similar. In this study, we found a significant increase in pro-inflammatory factors (NOS2, TNF-α and CXCL2) and no significant difference in three anti-inflammatory factors. We hypothesized that 50 μM AFB1-induced production of ROS promoted the secretion of pro-inflammatory cytokines and induced M1 polarization at least at the 24 h time point. However, it is curious that we found a significant decrease in expression of the M1 marker (CD86), as well as numerical decrease (not significant) in the M2 marker (CD206). 

The primary function of mitochondria is to generate energy used for cellular activities [50,51]. However, in the process of ATP production, a small percentage of electrons escape from the mitochondrial space, thus resulting in the production of ROS. Actually, mitochondria seem to be the most important sources of ROS production, generating almost 90% of the total cellular ROS [52]. In a mitochondrion, at least eight sites are known to be involved in the production of reactive oxygen species. However, mitochondrial Complexes I, II and III are thought to be the major contributors to the generation of excessive ROS [53,54]. Mitochondrial Complex I, which is also known as NADH dehydrogenase, consists of 45 subunits [55]. In our experiment, 50 μM AFB1 treatment elicited the upregulation of six genes of the ND gene family and downregulation of seven genes of the Nduf gene family. ND genes belong to the mitochondrial genome, and their products form the membrane arm of Complex I, which plays an important role in transferring protons [56,57]. The upregulation of ND genes might suggest that more protons were transferred from the mitochondrial matrix to intermembrane space, leading to higher membrane potential. In fact, a recent study reported that high membrane potential was beneficial for the production of superoxide [58]. On the contrary, Nduf genes belong to the nuclear genome, and their products form the matrix arm of Complex I, which serves as the electron acceptor from NADH and electron transfer pathways [57,59]. Therefore, the downregulation of Nduf genes will affect the structure and function of Complex I, causing more electrons escaping from the respiratory chain to oxygen forming superoxide. The products of SDHB and SDHD which were downregulated by AFB1 are subunits of Complex II. Complex II only transfers two electrons to the ubiquinone-bound membrane and does not contribute to the formation of a proton gradient [55]. Studies have suggested that a loss of SDHB, SDHC and SDHD would allow for the acceptance of an electron, but not progression along the respiratory chain, and consequently may enhance the production of ROS [60,61]. For Complex III, we found an upregulation of Cyt 1 and ISP and a downregulation of Cyt b. We speculate that the downregulation of Cyt 1 and ISP might affect the translocation of electrons, which leads to an accumulation in cytochrome b. These electrons would then be transferred to oxygen forming superoxide. This could also provide an explanation for the higher expression of Cyt b. In fact, the superoxide production in cytochrome b has been found in various studies [62,63]. Our results also show that the expression of COX-2 in Complex IV in the AFB1 group was significantly higher than in the control group. COX2 is an inducible protein that is constitutively expressed [64]. A previous study demonstrates that COX2 can increase ROS generation via its activity as a peroxidase to produce free radicals in the cytoplasm [65].

ROS has a dual function in cells. On the one hand, the strong oxidative properties of ROS can alter the structure and consequently the function of intracellular constituents such as proteins, lipids and nucleic acids. On the other hand, ROS is able to serve as an important short-lived second messenger in cell signaling pathway [66,67]. In our experiment, KEGG enrichment analysis found six signaling pathways in association with ROS and inflammatory response in macrophages, including PPAR, NF-κB, PI3K-Akt, mTOR and MAPK signaling pathways. NF-κB, as an important activator of inflammatory processes, is capable of regulating expression levels of inflammatory cytokines, chemokines and mediators of various cell types [68,69]. In fact, many studies have demonstrated that NF-κB could be activated directly by H_2_O_2_ which can induce strong oxidative stress in cells [70,71]. ROS can also activate some serine/threonine phosphorylation processes, and regulation of the PI3K-Akt signaling pathway has been reported to be related to ROS [72,73]. ROS can indirectly maintain the activation of PI3K through inhibition of its negative regulator, PTEN [74]. mTOR is located downstream of the PI3K-Akt signaling pathway and is recognized as the central kinase that controls cell proliferation and translation [75]. Recently, more evidence shows that ROS participate in the activation of mTOR [76,77]. Besides PI3K-Akt, the MAPK signaling pathway is also redox-sensitive and regulates a variety of cellular processes such as cell differentiation, survival and proliferation [72]. Endogenous ROS activates apoptosis signal regulated kinase 1 (ASK1) and promotes p38 MAPK signaling cascades in various cell types such as cardiac myoblasts and fibroblasts [74]. Although previous studies have confirmed that some pathways, such as NF-κB and MAPK signaling pathway [19,20], could be activated by AFB1, it is unclear whether other mentioned pathways, e.g., the PPAR, PI3K-Akt and mTOR signaling pathways, also play an important role in AFB1-induced oxidative stress and inflammatory response, and further research needs to be done.

## 4. Conclusions

AFB1 induces oxidative stress in macrophages via affecting the respiratory chain, which leads to the activation of several signaling pathways related to inflammatory response. Furthermore, we identified 25 genes which were relative with the generation of ROS after AFB1 exposure. Besides, seven signaling pathways associated with inflammatory response might be activated after ROS overproduction.

## 5. Materials and Methods

### 5.1. AFB1 Solutions Preparation

AFB1 was purchased from Pribolab (Pribolab, Qingdao, China). The AFB1 powder was dissolved in dimethyl sulfoxide (DMSO, Sigma, St. Louis, MO, USA) to make a stock solution (100 mM) that was stored at −20 °C in the dark. Different working solutions (0, 3.125, 6.25, 12.5, 25, 50 and 100 μM) were prepared using culture medium by two-fold dilution, and the final concentration of DMSO in the culture medium was 0.1% through all experiments, which did not cause any side effects on the cell line (data not shown). Before conducting the treatment experiments, the concentrations of working solutions were tested by high-performance liquid chromatography (HPLC; Shimadzu, Kyoto, Japan).

### 5.2. Cell Culture and Treatment

The RAW264.7 cell line was obtained from American type culture collection (ATCC, Manassas, VA, USA), and the cells were cultured by DMEM/F12 (Gibco, Carlsbad, CA, USA) supplemented with 10% fetal bovine serum (FBS, Hyclone, Logan, UT, USA) as well as 1% antibiotic/antimycotic (penicillin and streptomycin, Sigma, St. Louis, MO, USA) in cell culture flasks (Greiner, Vilvoorde, Gamany). Cells incubated at 37 ℃ in a humidified atmosphere of 5% CO2 incubator (Panasonic, Osaka, Japan).

### 5.3. Cell Viability Assay

To assess the toxic effect of AFB1 on the RAW264.7 cell line, cell viability was determined using Cell Counting Kit-8 (CCK-8, Dojindo, Japan). Cells were cultured in 96-well plates at a density of 1 × 10^4^ cells per well. The cells were incubated for 12 h and then treated with different concentrations of AFB1 (0, 3.125, 6.25, 12.5, 25, 50 and 100 μM) for 24 or 48 h. After treatment, CCK-8 was used to evaluate the cell viability according to the manufacturer’s instructions. The absorbance was measured at 450 nm using a Microplate Reader (Thermo Fisher, Waltham, MA, USA). In each group, six replicates were performed.

### 5.4. Determination of Intracellular Reactive Oxygen Species

The level of intracellular ROS in RAW264.7 cells was evaluated using 2,7-dichlorofluorescein diacetate (DCFH-DA; Sigma, St. Louis, MO, USA). Cells were cultured in 6-well plates at a density of 1 × 10^6^ cells per well. The cells were incubated for 12 h and then treated with different concentrations of AFB1 (0, 25, 50 and 100 μM) for 24 or 48 h. After treatment, the culture medium was removed and the cells were washed with PBS. Then, the cells were incubated with DCFH-DA diluted with DMEM/F12 for 20 min at 37 °C. Next, the slides were washed three times with PBS and scanned by laser scanning confocal microscopy using a Zeiss LSM700 META confocal system (Oberkochen, Germany), and cells assessed for green fluorescence. 

### 5.5. Determination of Glutathione and Malondialdehyde

The amount of glutathione (GSH) in RAW264.7 cells was evaluated using 5,5-dithiobis-(2-nitrobenzoic acid) (DTNB). Cells were cultured in 6-well plates at a density of 1 × 10^6^ cells per well. The cells were incubated for 12 h and then treated with different concentrations of AFB1 (0, 25, 50 and 100 μM) for 24 or 48 h. After treatment, the cells were harvested into 500 μL PBS and sonicated (Sonics VCX105, Oklahoma, OH, USA). GSH contents in the cell homogenate were determined at 405 nm by using a Reduced Glutathione assay kit (colorimetric method, Jiancheng, Nanjing, China) according to the manufacturer’s instructions, and total protein concentration was measured by using a BCA protein assay kit (Beyotime, Nantong, China).

The amount of malondialdehyde (MDA) in RAW264.7 cells was evaluated using thiobarbituric acid (TBA). Cells were cultured in 6-well plates at a density of 1 × 10^6^ cells per well. The cells were incubated for 12 h and then treated with different concentrations of AFB1 (0, 25, 50 and 100 μM) for 24 or 48 h. After treatment, the cells were harvested into 500 μL of PBS and repeatedly frozen and thawed for lysing cells. The cell homogenate was centrifuged at 12,000 rpm for 15 min at 4 °C. MDA contents in the supernatant were determined at 532 nm by using a Cell Malondialdehyde assay kit (colorimetric method, Jiancheng, Nanjing, JS, China) according to the manufacturer’s instructions, and total protein concentration was measured by using a BCA protein assay kit (Beyotime, Nantong, China).

### 5.6. Analysis of Inflammatory Cytokines Expression 

The mRNA levels of nitric oxide synthase 2 (NOS2), arginase 1 (ARG1), transforming growth factor-β (TGF-β), tumor necrosis factor-α (TNF-α), interleukin-6 (IL6), interleukin-10 (IL-10), chemokine (C-X-C motif) ligand 2 (CXCL2), cluster of differentiation 86 (CD86) and cluster of differentiation 206 (CD206) were detected by real-time quantitative PCR (RT-qPCR). Firstly, RAW264.7 cells were exposed to 0 or 50 μM AFB1 for 24 h. After treatment, cells were washed with PBS and harvested into 1 mL TRIzol (Invitrogen, Waltham, MA, USA) and total RNAs were extracted according to the manufacturer’s instructions. The quantities and qualities of RNAs were evaluated using a NanoPhotometer® N60/N50 (Implen, München, Germany). The complementary DNA (cDNA) was reverse-transcribed from 10 ng of total RNA using a ReverTra Ace^®^ qPCR RT Master Mix with gDNA Remover Kit (TOYOBO, Osaka, Japan). The levels of the transcripts of the target genes were determined using SYBR^®^ Green Realtime PCR Master Mix Kit (TOYOBO, Osaka, Japan) on an RT-qPCR system (BIO-RAD, Hercules, CA, USA). The relative amount of target mRNA was performed using the −2^−ΔΔCt^ method with GAPDH as a reference gene. The primers used in RT-qPCR reactions were synthesized by Sangon Biotech (Shanghai, China) and are showed in Table 3.

### 5.7. RNA-Seq and Bioinformatics Analysis 

RAW264.7 cells were cultured in 6-well plates at a density of 1 × 10^6^ cells per well. The cells were cultured for 12 h and then treated with medium with or without 50 μM AFB1 for 24 h. After treatment, cells were washed with PBS and harvested into 1 mL TRIzol (Invitrogen, USA) and total RNAs were extracted according to the manufacturer’s instructions. The qualities were evaluated by the ratio of OD260/OD280, and high-quality RNA was used for the construction of cDNA libraries. The cDNA libraries were sequenced using the Illumina HiSeq X10 platform. 

The raw sequence data were evaluated by FastQC and filtered by Trimmomatic-0.36 to remove low-quality sequences. After quality trimming, reads of >100 bp length with <2 mismatches were mapped to the *Mus musculus* genome using HISAT2 software. To calculate the gene expression and RPKM (reads per kilobase per million), StringTie and WGCNA tools were used. Differentially expressed genes (DEGs) between the control and AFB1 groups were detected using DESeq2. The lists of DEGs were filtered for those genes with a false discovery rate of *p* < 0.05 and an absolute fold-change cutoff of ≥2. 

Kyoto Encyclopedia of Genes and Genomes (KEGG) pathway enrichment analysis was carried out by ClusterProfiler to expound promising signaling pathways correlated with the overlapping DEGs. Adjusted *p* < 0.05 was defined as the cutoff criteria.

### 5.8. Statistical Analyses

All experiments were performed at least three times with similar results and the data obtained are expressed as the mean ± standard deviation (SD) unless otherwise specified. Analyses were performed using GraphPad Prism Version 7 software (GraphPad, San Diego, CA, USA). Statistical differences between control and the other groups were detected by t-tests and one-way analysis of variance (ANOVA) followed by multiple comparisons. Differences were considered statistically significant when the *p*-values were ≤ 0.05.

## Figures and Tables

**Figure 1 toxins-13-00401-f001:**
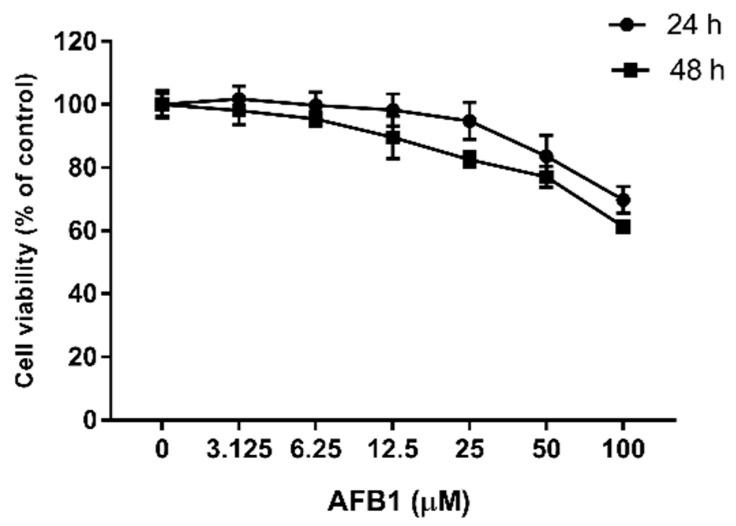
Dose-dependent toxicity of AFB1. RAW264.7 cells were exposed to different concentrations of AFB1 (0–100 μM) for 24 or 48 h, and then the viability was determined by CCK-8 assay. Data are expressed as mean values ± SD (*n* = 6).

**Figure 2 toxins-13-00401-f002:**
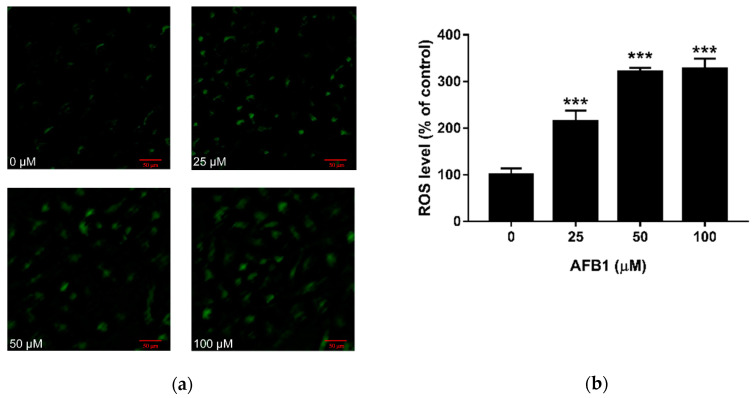
Effect of AFB1 on ROS production in RAW264.7 cells. Cells were exposed to different concentrations of AFB1 for 24 h. (**a**) Representative microscopy images of RAW264.7 cells treated with AFB1 are shown (100× magnification). (**b**) Quantitative analysis of green fluorescence intensity was calculated. Data are presented as mean values ± SD (*n* = 3). Significance compared with control, *** *p* < 0.001, scale bar = 50 μm.

**Figure 3 toxins-13-00401-f003:**
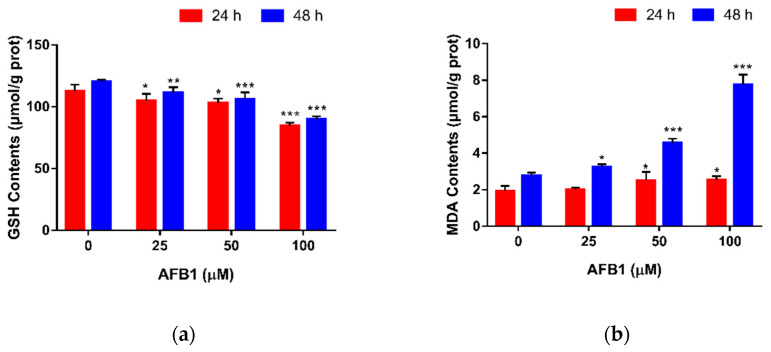
Effect of AFB1 on GSH and MDA contents in RAW264.7 cells. Cells were exposed to different concentrations of AFB1 for 24 or 48 h. The GSH (**a**) and MDA (**b**) contents in cells were determined as described in Section 5.5. Data are presented as mean values ± SD (*n* = 3). Significance compared with control, * *p* < 0.05, ** *p* < 0.01 and *** *p* < 0.001.

**Figure 4 toxins-13-00401-f004:**
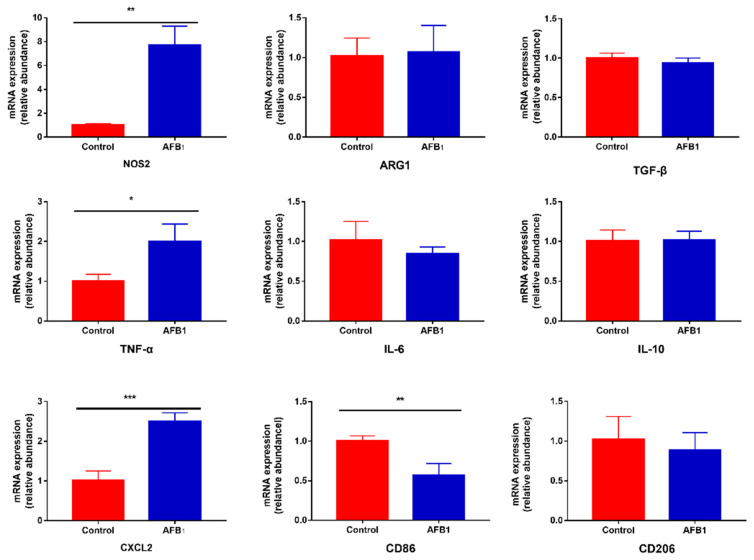
Effect of AFB1 exposure on gene expression of inflammatory cytokines. Cells were exposed to 50 μM AFB1 for 24 h. The expression levels of related genes were determined as described in Section 5.6. Data are presented as mean values ± SD (*n* = 3). Significance compared with control, * *p* < 0.05, ** *p* < 0.01 and *** *p* < 0.001.

**Figure 5 toxins-13-00401-f005:**
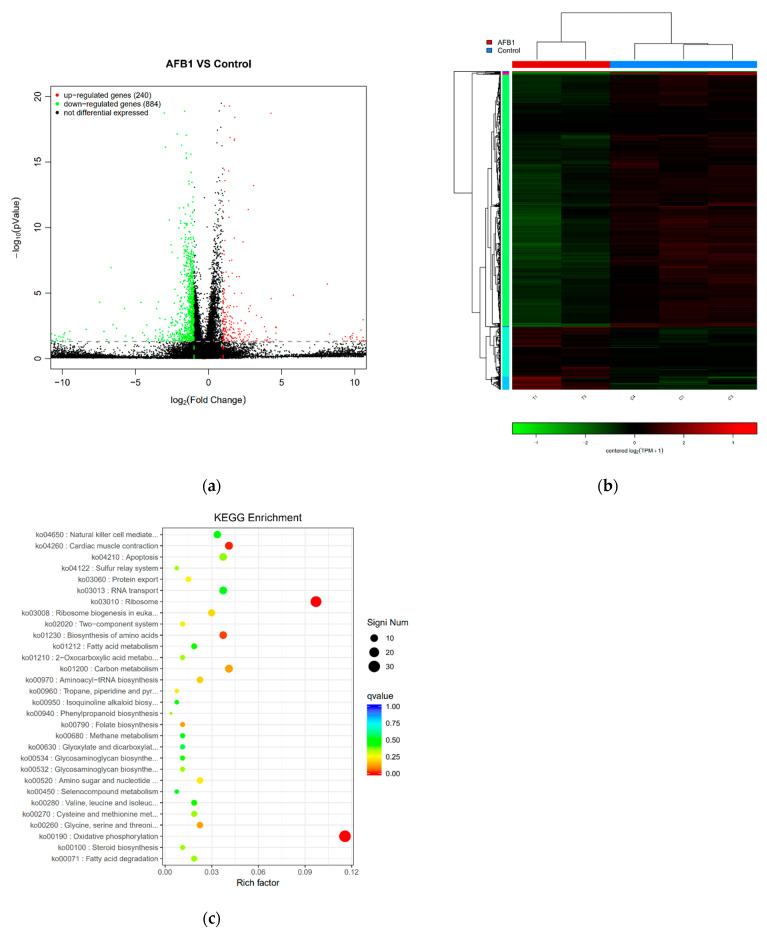
RNA-Seq analysis of the effect of AFB1 on RAW264.7 cells. (**a**) Volcano plots depicting DEGs were measured by RNA-Seq analysis. DEGs were defined by a magnitude fold change ≥ 2 and *p*-value < 0.05. (**b**) Heat map hierarchical clustering revealed genes that were differentially expressed in the 50 μM AFB1 group compared with the control groups. Red and green colors indicate higher expression and lower expression, respectively. (**c**) Scatter diagram of KEGG significant pathway enrichment. The color of the points represents the size of Q-value. The size of the points represents the number of DEGs contained in the pathways. The top 30 pathways ranked by *p*-value are shown.

**Figure 6 toxins-13-00401-f006:**
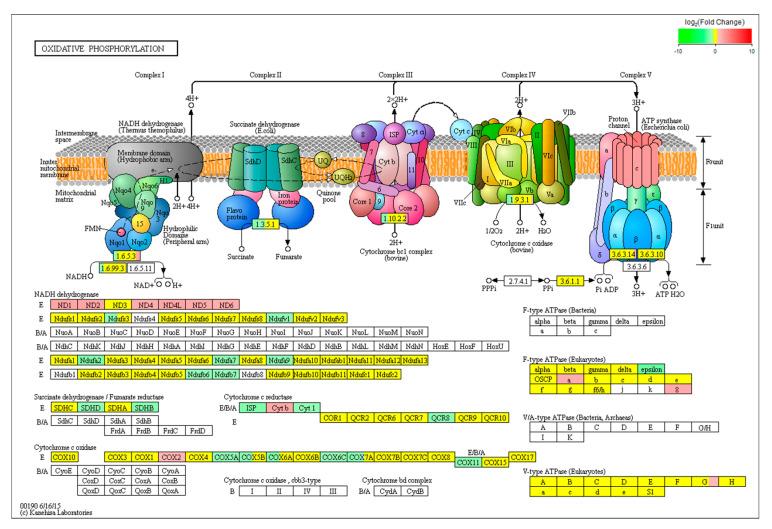
KEGG analysis of the effect of AFB1 on oxidative phosphorylation pathway. The oxidative phosphorylation pathway model was obtained from KEGG pathway. The green box indicates a gene that was downregulated by AFB1 treatment. The red box indicates a gene that was upregulated by AFB1 treatment. The yellow box indicates a gene that did not change significantly. The red box indicates a gene that was not detected.

**Table 1 toxins-13-00401-t001:** Sequencing analysis of inflammatory cytokines expression after AFB1 stimulation.

Genes ID	Gene Name	Abbreviations	Log_2_FC	*p*-Value	Regulation
ENSMUSG00000027398	Interleukin 1 beta	IL-1β	0.41	0.68	NS
ENSMUSG00000025746	Interleukin 6	IL-6	0.00	1.00	NS
ENSMUSG00000016529	Interleukin 10	IL-10	0.00	1.00	NS
ENSMUSG00000058427	Chemokine ligand 2	CXCL2	1.09	<0.001	UP
ENSMUSG00000024401	Tumor necrosis factor alpha	TNF-α	0.70	<0.001	UP
ENSMUSG00000020826	nitric oxide synthase 2	NOS2	−0.05	0.72	NS
ENSMUSG00000021253	Transforming growth factor, beta	TGF-β	−0.34	0.99	NS
ENSMUSG00000019987	Arginase 1	ARG1	−0.32	0.78	NS
ENSMUSG00000075122	CD80 antigen	CD80	0.30	<0.001	UP
ENSMUSG00000022901	CD86 antigen	CD86	−0.94	0.02	DOWN
ENSMUSG00000008845	CD163 antigen	CD163	−0.36	0.74	NS
ENSMUSG00000034783	CD206 antigen	CD206	−0.27	0.89	NS

**Table 2 toxins-13-00401-t002:** Expression dynamics of immune regulatory pathway mediated by oxidative stress after AFB1 stimulation.

KEGG ID	KEGG Term	*p*-Value	No. of DEGs	DEGs
ko03320	PPAR signaling pathway	0.1676	5	ACAA1B, FABP5, PLIN1, UCP1, CPT1C
ko04150	mTOR signaling pathway	0.2307	11	MAP2K2, LAMTOR5, STRADB, ATP6V1G2, FZD2, SLC3A2, SLC7A5, EIF4EBP1
ko04064	NF-κB signaling pathway	0.5093	4	BCL2A1B, BCL2A1A, RELB, IKBKG
ko04151	PI3K-Akt signaling pathway	0.7190	11	ITGB7, MAP2K2, LPAR1, COL2A1, LAMB3, EPHA2, CDKN1A, COL1A2, EIF4EBP1, IKBKG, OSM
ko04630	JAK-STAT signaling pathway	0.8721	4	SOCS6, IL12RB2, CDKN1A, OSM
ko04010	MAPK signaling pathway	0.9162	6	MAP2K2, GADD45A, FAS, DDIT3, RELB, IKBKG
ko04668	TNF signaling pathway	0.9344	4	CXCL2, FAS, IKBKG, PGAM5

The green word indicates a gene that was downregulated by AFB1 treatment. The red word indicates a gene that was upregulated by AFB1 treatment.

**Table 3 toxins-13-00401-t003:** Forward/Reverse primer sequences of RT-qPCR.

Genes	Primer Position	Primer Sequence	Genebank Number
GAPDH	Forward	AGGTCGGTGTGAACGGATTTG	GU214026.1
	Reverse	TGTAGACCATGTAGTTGAGGTCA	
NOS2	Forward	GTTCTCAGCCCAACAATACAAGA	AY090567.1
	Reverse	GTGGACGGGTCGATGTCAC	
ARG1	Forward	CCCGACTTCTGGGACTTCTG	AB047402.1
	Reverse	AGTAGGTTCCGAAGACTGGGT	
TNF-α	Forward	CCCTCACACTCAGATCATCTTCT	NM_013693.3
	Reverse	GCTACGACGTGGGCTACAG	
TGF-β	Forward	CTCCCGTGGCTTCTAGTGC	NM_011577.2
	Reverse	GCCTTAGTTTGGACAGGATCTG	
IL-6	Forward	TAGTCCTTCCTACCCCAATTTCC	NM_031168.2
	Reverse	TTGGTCCTTAGCCACTCCTTC	
IL-10	Forward	GCTCTTACTGACTGGCATGAG	NM_010548.2
	Reverse	CGCAGCTCTAGGAGCATGTG	
CXCL2	Forward	CCAACCACCAGGCTACAGG	NM_008625.2
	Reverse	GCGTCACACTCAAGCTCTG	
CD86	Forward	TGTTTCCGTGGAGACGCAAG	NM_019388.3
	Reverse	TTGAGCCTTTGTAAATGGGCA	
CD206	Forward	CTCTGTTCAGCTATTGGACGC	NM_008625.2
	Reverse	CGGAATTTCTGGGATTCAGCTTC	

## Data Availability

Data are available upon request, please contact the contributing authors.

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
