# Peer review of "Transcriptional Profiling of Aflatoxin B1-Induced Oxidative Stress and Inflammatory Response in Macrophages"

_toxins, 2021, doi:10.3390/toxins13060401_

Round 1
Reviewer 1 Report
The article is very interesting, involving a huge amount of work. I've identified few spelling mistakes. A thorough check by a native English speaker could be very usefull.
Corrections suggested
-line 13 decreased level glutathione level- delete first word level
-line 17 Taken together ..replace with Alltogether
-line 44 On one hand could be deleted or replaced with Firstly,
line 46-In vitro =in vitro
line 103 AFB1on= AFB1 on
line 162 reproducible= reproductible
line 297 A recent study= a recent study
line 420 1×106 cells = 1×106 cells
Reviewer 2 Report
Manuscript Number: toxins-1230021
Title: Transcriptional profiling of aflatoxin B1-induced oxidative stress and inflammatory response in macrophages
Type: Article
This manuscript aimed to investigate that aflatoxin B1 could induce oxidative stress in macrophages via affecting the respiratory chain, which leads to the activation of several signalling pathways related to the inflammatory response. I believe that this manuscript is a good contribution to research in this area. The objectives of this manuscript were carried and are significant for further research.
Comments:
The subject of the manuscript is consistent with the scope of the journal Toxins. There are no errors of fact or logic. The abstract does bring out the main points of the paper. The literature references are adequate and recent. The design of the study and the methodology: e.g. cell viability assay, determination of intracellular reactive oxygen species, determination of glutathione and malondialdehyde, analysis of inflammatory cytokines expression and RNA-seq and bioinformatics analysis are appropriate, processed clearly and concisely.
1.
Page 3, line 98 and 112
Page 4, line 130
Page 5, line 147
Page 6, line 174
Page 7, line 197
The figures are of low quality and difficult to read.
I don´t have cardinal objections.
So, the manuscript is fit for possible publication in TOXINS journal (accept after minor revision)!
